chemical physics/crystallography

particle morphology, structure, properties, nascent ultra-high molecular weight polyethylene

**Author for correspondence:**
Xinwei Wang
e-mail: 56537858@qq.com

This article has been edited by the Royal Society of Chemistry, including the commissioning, peer review process and editorial aspects up to the point of acceptance.

# Particle morphology, structure and properties of nascent ultra-high molecular weight polyethylene

Wenyang Zhang[1], Zhengwen Wu[1], Hanjun Mao[1], Xinwei Wang[1], Jianlong Li[1], Yongyi Mai[1] and Jianyong Yu[2]

[1]State Key Laboratory of Polyolefins and Catalysis, Shanghai Key Laboratory of Catalysis Technology for Polyolefins, Shanghai Research Institute of Chemical Industry, Shanghai 200062, People's Republic of China
[2]School of Materials Science and Engineering, Donghua University, Shanghai 201620, People's Republic of China

(iD) WZ, 0000-0002-8761-9361

The effects of particle morphology on the structure and swelling/dissolution and rheological properties of nascent ultra-high molecular weight polyethylene (UHMWPE) in liquid paraffin (LP) were elaborately explored in this article. Nascent UHMWPE with different particle morphologies was prepared via pre-polymerization technique and direct polymerization. The melting temperature and crystallinity of UHMWPE resins with different particle morphologies were compared, and a schematic diagram was proposed to illustrate the mechanism of UHMWPE particle growth synthesized by pre-polymerization method and direct polymerization. The polymer globules in the nascent UHMWPE prepared by using pre-polymerization technique are densely packed and a positive correlation between the particle size and the viscosity-averaged molecular weight can be observed. The split phenomenon of particles and the fluctuation in the viscosity of UHMWPE/LP system prepared by direct polymerization can be observed at a low heating rate and there is no correlation between particle size and viscosity-averaged molecular weight.

## 1. Introduction

Controlling the morphology of nascent polymer particle in modern polyolefin industry by manipulating the catalyst and reaction conditions has aroused widespread concern both in academia and industry. Uncontrolled polymer morphology may cause serious

fouling, sheeting in the polymerization reactor, leading to reactor shutdown and production loss [1]. A good particle morphology from an industrial perspective usually means spherical shape, narrow particle size distribution, high bulk density, controlled degree of porosity and internal composition, etc. [2,3]. Several models have been proposed over the years to elucidate the particle growth mechanism, among which the solid core model, flow model and the multigrain model [4–6] are the most popular and widely used. According to the solid core model, polymerization reaction occurs at the surface of a non-friable catalyst sphere. And in the flow model, the growing polymer continuously flows outward, and polymer accumulates outside the particle. Based on the multigrain model, the catalyst grain is composed of microparticles, and polymerization occurs on the surface of the microparticles, forming polymer layer around them and causing the particle to expand progressively as polymerization proceeds.

The particle morphology of polyolefins has been generally influenced by the morphologies of catalyst particles, polymerization condition and the effect of molecular organization such as polymer crystallization [4,7]. In olefin polymerization, the catalyst particle morphology such as shape, pore and particle size distribution [8–13] has an important impact on the rate of mass transport of reactants and the diffusion of diluents inside the growing particle, and thus it affects the rate of polymerization and molecular architecture of final polymer including chain length distribution, comonomer content and so on [14,15]. In addition, polymerization conditions, especially in the initial stages of polymerization, also play a key role in controlling the particle morphology. Previous studies [2,14,15] have proved that using pre-polymerization technique, polymerization occurs under milder condition, and the initial particle growth could be better controlled, which influences the final particle morphology [16]. Besides, the polymer microstructure is strongly dependent on the interaction between polymerization and crystallization over time [2], which in turn affect the morphology of polymer particles. Thus, the properties of the final polymer product depend not only on the polymer structure including molecular architecture and crystallization, but also on the morphology of polymer particles.

Ultra-high molecular weight polyethylene (UHMWPE), a typical polyolefin having a viscosity-average molecular larger than $1 \times 10^6$ g mol$^{-1}$, is featured by excellent self-lubrication properties, high impact strength and high chemical stability, which is widely used in many applications including textile, chemical industry, packaging, artificial joints and other field [16–19]. However, the polymer chains of UHMWPE are prone to become entangled, forming a large number of physical entanglements. And the dense physical entanglements and enhanced intermolecular interactions bring great challenges for its processing due to its high melt viscosity and poor melt flowability [20].

To improve the processing performance, one effective way is to dilute the UHMWPE with flow modifiers, such as decalin, liquid paraffin (LP), etc. In order to form a homogeneous solution, UHMWPE nascent powders are required to be dissolved in petrochemical solvent, involving the swelling and dissolution of polymer. The dissolution of amorphous polymer in the solvent involves two transport processes, namely solvent diffusion and chain disentanglement [21]. The mixing process between an amorphous polymer and a solvent would occur spontaneously when the value of the free energy change on mixing ($\Delta G_m$) is negative, according to equation [22]

$$\Delta G_m = \Delta H_m - T\Delta S_m, \tag{1.1}$$

where $\Delta H_m$ is the enthalpy change on mixing, $T$ is the absolute temperature and $\Delta S_m$ is the entropy change on mixing. Partial swelling could occur for the cross-linked polymers, and the dissolution of crystalline polymers in good solvents is closely related to the melting of crystalline regions [23]. As a highly crystalline polymer, the swelling and dissolution of UHMWPE resin are bound to be associated with the melting of crystals related to the polymer crystallization in olefin polymerization, heating rate and diluent content. In this study, nascent UHMWPE with different particle morphologies was synthesized via pre-polymerization technique or direct polymerization, and the differences in the particle morphology, structure, swelling/dissolution and rheological properties in LP were investigated by scanning electron microscope (SEM), differential scanning calorimetry (DSC), thermal microscope and rotary rheometer.

# 2. Experimental section

## 2.1. Preparation of nascent UHMWPE

A high activity, spherical TiCl$_4$/MgCl$_2$ catalyst (donated by the Shanghai Research Institute of Chemical Industry) was used in the polymerization reactions in the presence of hydrogen, with Al(C$_2$H$_5$)$_3$ as a

cocatalyst. After purification and evacuation of the reactor system, it was subsequently filled with solvent $n$-hexane, $TiCl_4/MgCl_2$ catalyst, $Al(C_2H_5)_3$ cocatalyst and a small amount of hydrogen. The reaction mixture was directly heated to 70°C in a 10 l reaction kettle with an impeller speed of 400 r.p.m. and the ethylene pressure was kept constant at 0.7 MPa, and then ethylene slurry polymerization was started (without the pre-polymerization stage). Another improved polymerization process is applied by using pre-polymerization technique. A slow pre-polymerization was conducted at 30°C for 30 min with the reactor pressure of 0.1 MPa. Subsequently, the polymerization reaction was carried out at 70°C with the reactor pressure of 0.7 MPa until the end of polymerization.

## 2.2. Polymer characterization

The intrinsic viscosity [$\eta$] of UHMWPE in decahydronaphthalene was measured using an Ubbelohde viscometer at 135°C according to ISO 1628-3. The viscosity-averaged molecular weight ($M_\eta$) was calculated according to the equation $M_\eta = 53\,700 \times [\eta]^{1.49}$.

The particle size distributions of UHMWPE powders were conducted on a Microtrac S3500 laser particle size analyser. Particle size distribution is defined by $(D90–D10)/D50$, where the $D10$, $D50$ and $D90$ values represent the particle diameter corresponding to the 10%, 50% and 90% cumulative percentage.

The XRD measurements were performed at room temperature using the X-ray diffractometer Rigaku Ultima IV (Japan) with a Cu–K$\alpha$ source ($\lambda = 1.54$ Å, 40 kV and 200 mA). The measured angle $2\theta$ ranged from 5 to 40° with a scan speed of 3° min$^{-1}$. The XRD profiles were fitted by several crystalline peaks and one amorphous halo. The crystallinity ($X_c$) was calculated by the ratio of the areas of the crystalline peaks to the total areas.

The melting behaviour were conducted using a differential scanning calorimeter (Netzsch DSC 204F1 Phoenix, Selb, Germany) at a heating rate of 1 or 10°C min$^{-1}$ with temperature ranging from 30 to 210°C. The crystallinity was calculated from the following equation:

$$X_c = \frac{\Delta H}{\Delta H_m^0} \times 100\%,  \tag{2.1}$$

where $\Delta H$ is the melting enthalpy of nascent UHMWPE and $\Delta H_m^0$ is the melting enthalpy of 100% crystalline PE (293 J g$^{-1}$) [24].

The morphology and microstructure of nascent UHMWPE particles were characterized by a field emission scanning electron microscope (FE-SEM, ZEISS Merlin Compact, Germany).

Optical photographs of UHMWPE particles in LP during the heating process were observed by a Linkam THMS 600 stage with imaging station (Linkam Scientific Instruments, UK).

The viscosities of UHMWPE/LP system at designed concentrations with temperature were measured by using a rheometer (MCR 302, Anton Paar, Austria). The heating rate is 1 or 10°C min$^{-1}$ at a rotary speed of 300 r.p.m.

PE-140 was synthesized via pre-polymerization technique and PE-120 was prepared by direct polymerization.

# 3. Results and discussion

## 3.1. The morphology of nascent UHMWPE

Figure 1 shows the SEM micrographs of PE-140 and PE-120 at different magnifications. At a magnification of ×300, the particles are approximately spherical in shape for PE-140, while the particles show the irregular shape for PE-120, suggesting that using pre-polymerization technique is an effective way to achieve good particle morphology. At a magnification of ×1000, most of the nascent particles for PE-140 have the spherical shape, replicating the shape of the catalyst; however, a poor replication of the original catalyst particle shape for PE-120 can be observed. Furthermore, the spherical particles comprising many polymer globules consisting of numbers of smaller globules can be observed for both PE-120 and PE-140, indicating the catalyst fragmentation during polymerization and in contrast with PE-140, the polymer globules in the PE-120 particles are not densely packed. At a magnification of ×5000, it is also clear that the polymer globules are held together by the polymer fibrils as marked by the red line. Compared with PE-140, a larger number of fibrils can be seen for PE-120. It has been reported that the polymer fibrils are the product of intermolecular crystallization

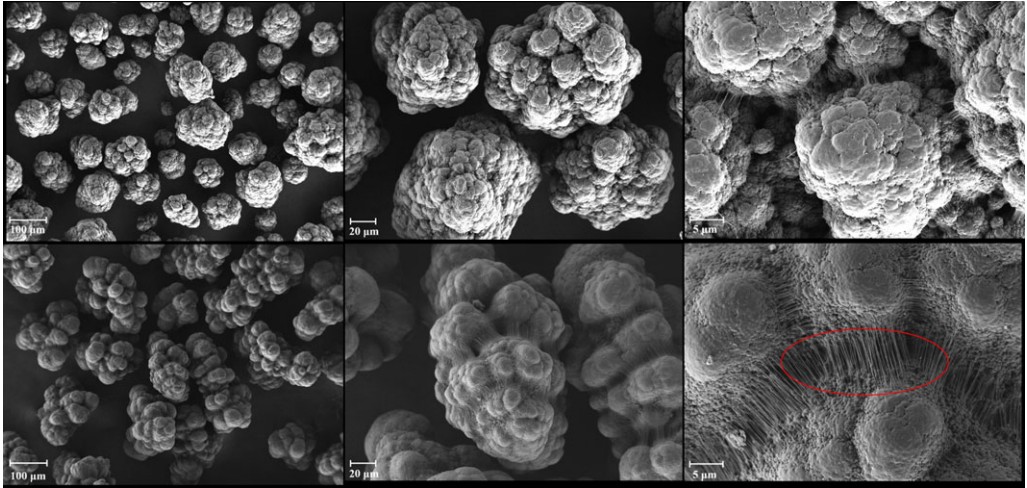

**Figure 1.** SEM micrographs of PE-140 (top) and PE-120 (bottom) with three different magnifications of ×300 (left), ×1000 (middle) and ×5000 (right).

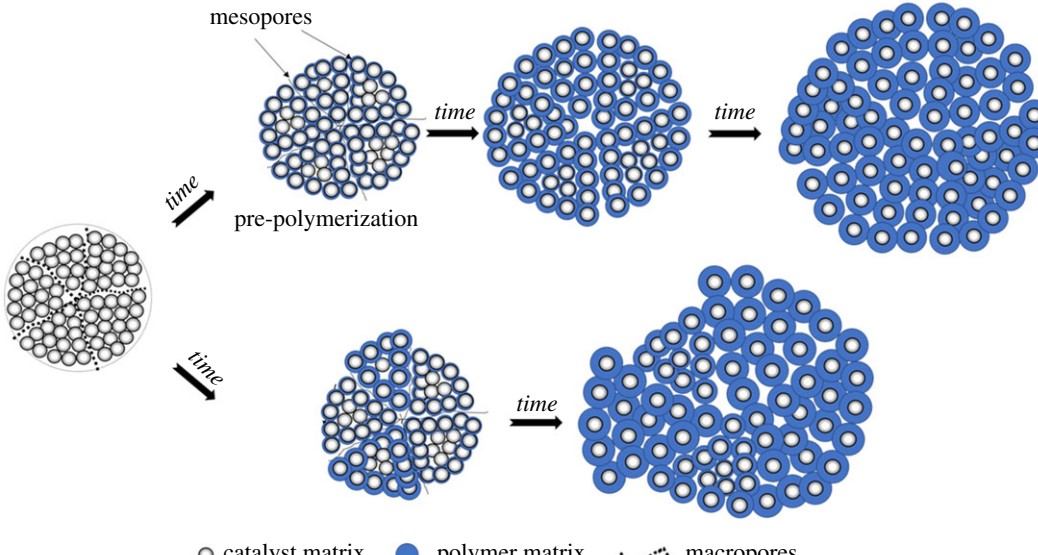

**Figure 2.** The schematic diagrams on the mechanism of UHMWPE particle growth synthesized by pre-polymerization method (top) and direct polymerization (bottom).

of polymer chains (extended-chain crystals), and the structure unit of the polymer globule is closely related to the intramolecular crystallization of polymer chains (chain-folded crystals) [25,26].

Based on the above result and discussion, the schematic diagrams on the mechanism of polymer growth are illustrated in figure 2 on the basis of the multigrain model (MGM) [5,6]. Once injected into the reaction kettle, the monomers in solvent *n*-hexane diffuse through the surface of catalyst particles into the macropores inside the catalyst. As soon as the ethylene monomers reach the active sites, polymerization occurs under milder conditions via pre-polymerization technique and the polymer layer over the active sites could be formed until a limiting value in pore size is reached. Then, the catalyst grain is broken first on the largest macropores, then on successively smaller pores for a short time until the grain is broken into relatively 'homogeneous' microparticles. Thus, relatively uniform polymer shells could be formed via pre-polymerization technique. It should be noted that the thin polymer layer over the active sites is porous and loose, and the monomers could easily diffuse into the interior of catalyst particles through the polymer shell in the subsequent polymerization. As polymerization proceeds, further fragmentation begins at the outermost layer and progresses towards the centre for catalyst microparticles, forming smaller units fragments and the main polymerization takes place. For the polymerization reaction without pre-polymerization, heterogeneous polymerization could occur due to the different pore sizes inside the

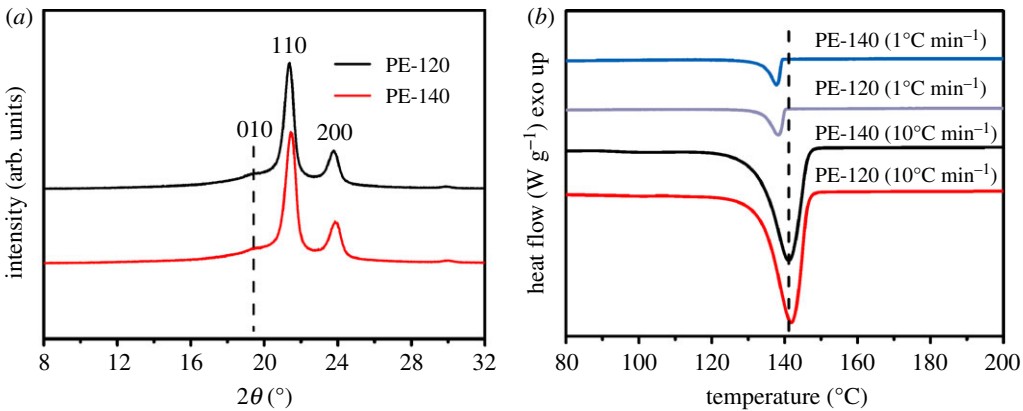

**Figure 3.** (a) Crystal structure and (b) DSC heating curves at different heating rates of PE-140 and PE-120 (exo up: upward peaks represent exothermic events, while downward peaks are characteristic of endothermic events).

**Table 1.** Physical properties of UHMWPE resins.

| sample | $M_\eta$ (g mol$^{-1}$) | particle size (µm) | particle size distribution | bulk density (g cm$^{-3}$) |
|---|---|---|---|---|
| PE-140 | $140 \times 10^4$ | 115.9 | 0.68 | 0.50 |
| PE-120 | $120 \times 10^4$ | 178.9 | 0.78 | 0.45 |

catalyst particles, leading to uneven fragmentation and the loss of control over the particle morphology in the early stage of polymerization reaction. In addition, the increased diffusion resistance to monomer penetration into the interior of catalyst particles through the thick polymer shell synthesized by rapid polymerization might lead to the mass transfer limitations and the particle overheating due to the violent reaction might lead to the partial deactivation of catalytic activity, causing a decrease in polymerization rate in the subsequent polymerization. Thus, a good particle morphology with spherical shape, narrow particle size distribution and high bulk density could be obtained by using pre-polymerization technique, as presented in table 1.

As mentioned in Introduction, the properties of polymer depend not only on the polymer structure including molecular architecture and crystallization, but also on the morphology of polymer particles. To further explore the effect of particle morphology on the properties of polymer resins, thermal microscope, DSC and rotary rheometer were used to elaborate the swelling/dissolution and rheological behaviours of UHMWPE in LP.

## 3.2. Crystal structure and melting behaviour

Figure 3a displays the crystal structure of PE-140 and PE-120. Two distinct diffraction peaks: (110) at $2\theta = 21.3°$ and (200) at $2\theta = 23.8°$ are observed for both PE-140 and PE-120, corresponding to the orthorhombic phase of PE [27]. Besides, a weak shoulder peak at around $2\theta = 19.5°$ can be assigned to the 010 diffraction of monoclinic phase of PE [28]. It has been reported that the monoclinic phase is metastable and could be obtained from the orthorhombic phase in the presence of mechanical stress [29]. As mentioned above, the catalyst grain is broken into relatively 'homogeneous' microparticles in the early stage of reaction, generating some deep cracks. And the polymer chains in the cracks are stretched by mechanical stress, leading to the formation of extended-chain crystals under appropriate conditions, and as polymerization proceeds, the clusters will produce extended-chain polyethylene fibrils. The fibril structure stretched by mechanical stress is prone to produce the monoclinic phase, forming the diffraction peak of (010) crystal plane. Figure 3b depicts the DSC heating curves at different heating rates of PE-140 and PE-120. The melting peaks gradually move to a higher temperature for both PE-140 and PE-120 as the heating rate increases, which is attributed to superheating as well as thermal lag [30,31].

The melting temperature ($T_m$), the onset of melting temperature ($T_{m\ onset}$) and the crystallinity are obtained by the melting curve with a heating rate of 10°C min$^{-1}$.

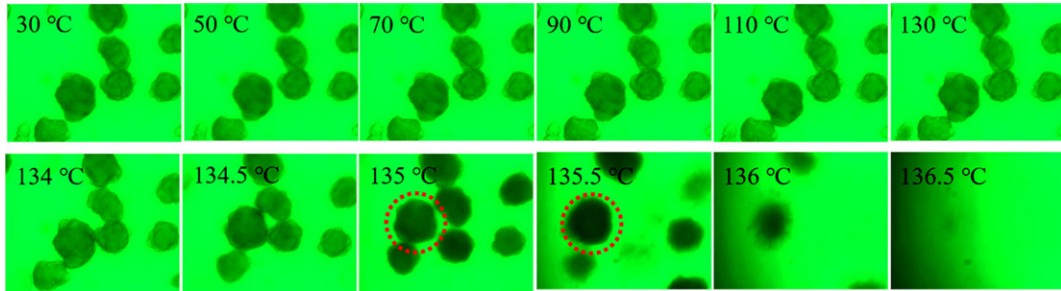

**Figure 4.** Selective optical photographs of PE-140 in LP at a heating rate of 1℃ min⁻¹.

**Table 2.** The melting temperature and crystallinity of nascent UHMWPE.

| sample | $T_m$ (°C) | $T_{m\ onset}$ (°C) | crystallinity (DSC) (%) | crystallinity (XRD) (%) |
|--------|-----------|---------------------|-------------------------|-------------------------|
| PE-140 | 141 | 133.1 | 65.4 | 47.2 |
| PE-120 | 142 | 133.6 | 71 | 50 |

Table 2 depicts the melting temperature and the crystallinity of PE-140 and PE-120. As can be seen, the melting temperature of PE-140 and PE-120 are 141°C and 142°C, respectively. And the onset of melting temperature of PE-140 is 133.1°C, slightly below the onset of melting temperature of PE-120 (133.6°C). The crystallinity of PE-140 is 65.4%, while the crystallinity of PE-120 is 71%. By contrast, the crystallinity was also calculated by the XRD profiles. A higher crystallinity for PE-120 (50%) compared with PE-140 (47.2%) could also be seen from table 2. As mentioned above, an increase in temperature due to violent reaction may lead to a decrease in the crystallization rate for PE-120, and the entangled polymer chains tend to untangle and orient under mechanical stress. And it seems much easier to crystallize for oriented polymer chains due to the decrease in conformational entropy, forming extended-chain crystals. Thus, the melting temperature and crystallinity of PE-120 are slightly higher than those of PE-140.

## 3.3. Swelling and dissolution behaviour

As a non-polar and highly crystalline polymer, the swelling and dissolution of UHMWPE resin in LP are bound to be closely associated with the melting behaviour. However, the melting behaviour of nascent UHMWPE is also related to the polymer crystallization in olefin polymerization, heating rate and diluent content. Thus, the swelling and dissolution behaviour of UHMWPE in LP were first investigated at different heating rates.

Figure 4 shows the selective optical photographs of PE-140 in LP at a heating rate of 1°C min⁻¹. The volume of UHMWPE particles stays almost constant with the temperature ranging from 30°C to 130°C. This can be interpreted that the volume of particle could not be enlarged due to the limitation of interpenetrating networks composed of interlocked crystallites and highly entangled amorphous phase. The colour of particles in LP deepens with an increase in temperature, especially at 135 and 135.5°C marked by the red dotted line, suggesting the crystals start to melt. Meanwhile, an increase in particle volume with temperature originating from the diffusional transport of solvent (LP) into the interior of particles can also be observed, indicating the networks are gradually destroyed due to the continuous melting of crystallites. As temperature continues to increase, the particles gradually fade from view until they are no longer visible at 136.5°C, suggesting the crystals are completely melted and the observed particles gradually become transparent. It is noteworthy that a homogeneous solution may not be obtained at 136.5°C because of the short diffusion times.

The selective optical photographs of PE-120 in LP at a heating rate of 1°C min⁻¹ are presented in figure 5. As with PE-140, the volume of PE-120 particles stays almost constant with the temperature ranging from 30 to 130°C, and a deep colour and an increase in particle volume also can be observed as temperature continues to increase. A major difference between PE-140 and PE-120 is the

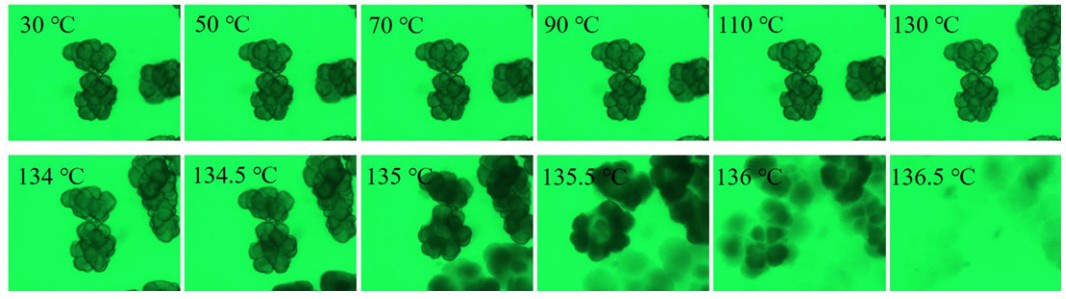

**Figure 5.** Selective optical photographs of PE-120 in LP at a heating rate of 1°C min$^{-1}$.

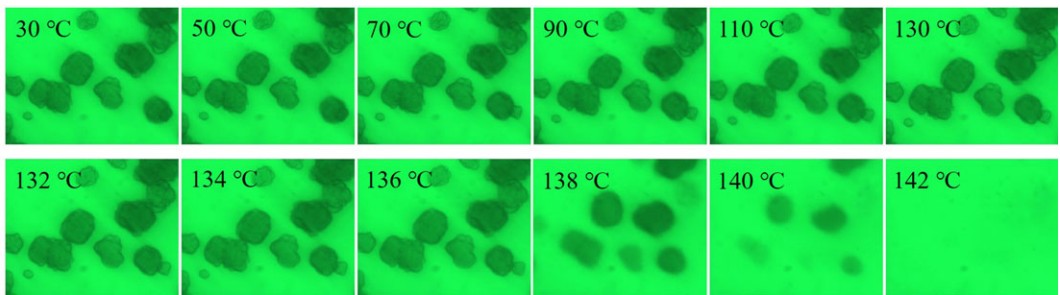

**Figure 6.** Selective optical photographs of PE-140 in LP at a heating rate of 10°C min$^{-1}$.

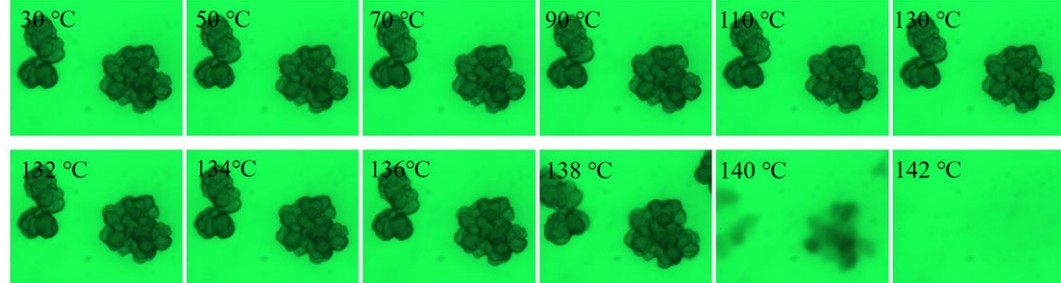

**Figure 7.** Selective optical photographs of PE-120 in LP at a heating rate of 10°C min$^{-1}$.

fragmentation of particles for PE-120 in LP. The polymer globules in the PE-120 particles are not densely packed, and held together by a large number of polymer fibrils as shown in SEM. The LP can easily penetrate into the fibrils between the polymer globules in the PE-120 particles. The fibrils are first melted due to the dilution effect of solvent on the melting point of semi-crystalline polymers, and the polymer globules gradually diffuse into LP, producing the split phenomenon of PE-120 particles.

Figures 6 and 7 depict the selective optical photographs of PE-140 and PE-120 in LP at a heating rate of 10°C min$^{-1}$. The volume of UHMWPE particles stays almost constant for both PE-140 and PE-120 with the temperature ranging from 30 to 130°C. As temperature continues to increase, the observed particles gradually fade until they are no longer visible at 142°C. This is because faster heating rate would lead to a rapid melting of crystals, and the swelling of particles are hardly observed until the crystals are completely melted.

## 3.4. Rheological behaviour

Subsequently, the effect of particle morphology on the rheological properties of nascent UHMWPE in LP with different concentrations (0.03 and 0.5%) was further investigated. Viscosity changes of UHMWPE/ LP system with the concentration of 0.03% at heating rates of 1°C min$^{-1}$ (figure 8$a$) and 10°C min$^{-1}$ (figure 8$b$) are illustrated in figure 8. The changes in viscosity over a wide temperature range are

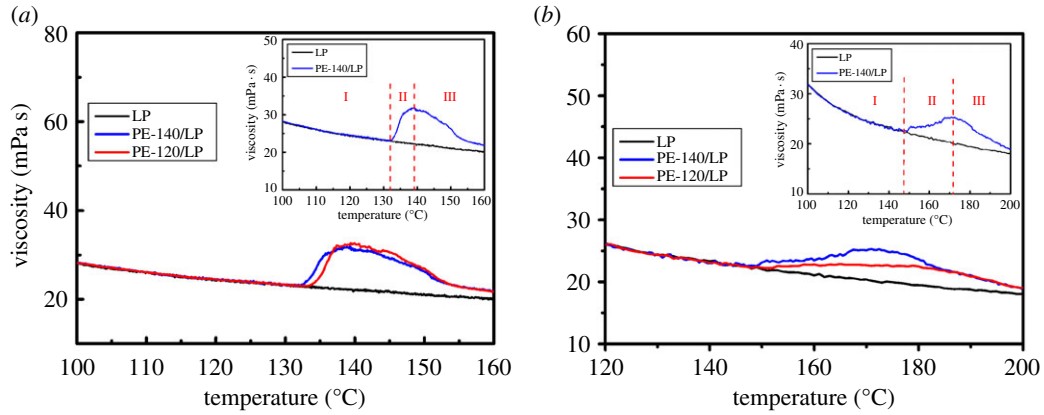

**Figure 8.** Viscosity changes of UHMWPE/LP system with the concentration of 0.03% at heating rates of 1℃ min⁻¹ (*a*) and 10℃ min⁻¹ (*b*).

almost the same under different heating rates. The onset temperature for PE-120 corresponding to the sudden increase in the viscosity is higher than the temperature for PE-140 under the same heating rates in accordance with the variation of melting temperature, suggesting the melting of crystals is a prerequisite for the swelling/dissolution of nascent UHMWPE in LP.

Combining the result of the swelling and dissolution behaviour, the evolution of viscosity of UHMWPE/LP system with temperature could be divided into three different stages: swelling equilibrium stage (region I), dissolution stage (region II) and rheological stage (region III), as shown in the inset in figure 8. In region I, the viscosity decreases gradually with temperature and only a small amount of LP diffuses into the interior of particles, reaching the swelling equilibrium state. And the viscosity of UHMWPE/LP system with increasing temperature is consistent with the viscosity of LP, suggesting the main contribution to the viscosity comes from the solvent (LP). As the temperature increases, the internal friction of molecules decreases due to the increased distance between the molecules caused by the enhanced thermal motion, so the viscosity of solvent (LP) decreases with temperature. In region II, as temperature continues to increase, the viscosity of UHMWPE/LP system increases dramatically and attains a maximum. At a heating rate of 1℃ min⁻¹ as illustrated in figure 8*a*, the temperature of UHMWPE/LP system gradually reaches the onset of melting temperature, and crystals start to melt. It should be noted that the onset of melting temperature of UHMWPE in LP is lower than the onset of melting temperature of nascent UHMWPE due to the dilution effect of solvent. And the melted polymer chains gradually diffuse into the solvent, leading to a continuous increase in the viscosity. However, at a heating rate of 10℃ min⁻¹ as shown in figure 8*b*, a rapid melting of crystals occurs and a continuous increase in the viscosity can be observed due to the completely melted polymer chains diffusion into the solvent. In region III, the viscosity of UHMWPE/LP system decreases sharply with increasing temperature. The polymer chains tend to be gradually oriented along the flow direction. At the same time, the thermal motion is enhanced with increasing temperature, resulting in a sharp decrease in the viscosity of UHMWPE/LP system.

Viscosity changes of UHMWPE/LP system with the concentration of 0.5% at heating rates of 1℃ min⁻¹ (figure 9*a*) and 10℃ min⁻¹ (figure 9*b*) are presented in figure 9. The evolution of viscosity with temperature could also be divided into three different stages: swelling equilibrium stage, dissolution stage and rheological stage. Different from the case at lower concentrations (0.03%), the fluctuation in the viscosity of UHMWPE/LP system marked by the black dotted line with the concentration of 0.5% during heating can be observed at a heating rate of 1℃ min⁻¹ as shown in figure 9*a*. Meanwhile, the onset temperature for PE-120 corresponding to the sudden increase in the viscosity is lower than the temperature for PE-140 at a heating rate of 1℃ min⁻¹. According to our previous analysis, the polymer globules in the PE-120 particles are not densely packed, and are held together by a large number of polymer fibrils. The fibrils are first melted due to the dilution effect of solvent and the melted polymer chains gradually diffuse into the solvent, causing the sudden increase in the viscosity to occur at a lower temperature compared with PE-140. Besides, the probability of colliding with other polymer globules connected by the melted polymer chains increases, leading to the fluctuation in the viscosity of UHMWPE/LP system during heating. However, at a heating rate of 10℃ min⁻¹ as presented in figure 9*b*, no obvious fluctuation can be observed due to the rapid melting of crystals.

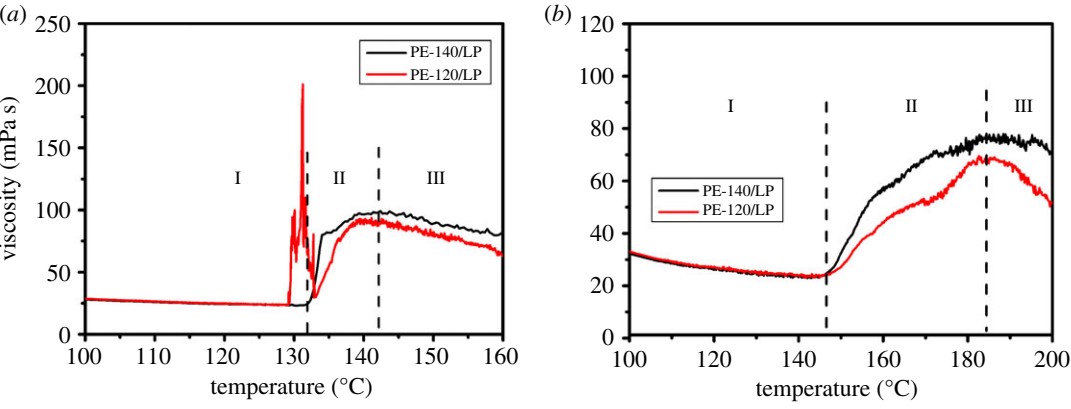

**Figure 9.** Viscosity changes of UHMWPE/LP system with the concentration of 0.5% at heating rates of 1°C min$^{-1}$ (a) and 10°C min$^{-1}$ (b).

**Table 3.** Particle size, particle size distribution and viscosity-averaged molecular weight of sieved UHMWPE.

| samples | | particle size (µm) | particle size distribution | $M_\eta$ (g mol$^{-1}$) |
|---|---|---|---|---|
| PE-140 | UHMWPE-l | 176 | 0.66 | $170 \times 10^4$ |
| | UHMWPE-m | 138.6 | 0.60 | $147 \times 10^4$ |
| | UHMWPE-s | 86.94 | 0.59 | $118 \times 10^4$ |
| PE-120 | UHMWPE-l | 208.1 | 0.90 | $110 \times 10^4$ |
| | UHMWPE-m | 158.6 | 0.68 | $127 \times 10^4$ |
| | UHMWPE-s | 138.8 | 0.72 | $121 \times 10^4$ |

The particle size and its distribution as an important index of resin affect the processing of UHMWPE, and thus, we further explore the differences in structure and properties of nascent UHMWPE with different particle sizes after sieving. The nascent UHMWPE is repeatedly sieved by use of standard sieves with different pore sizes. The sieved UHMWPE is named UHMWPE-l, UHMWPE-m and UHMWPE-s in order of particle size. Table 3 depicts the particle size, particle size distribution and viscosity-averaged molecular weight of sieved UHMWPE. It can be seen that UHMWPE particles with different particle sizes can be obtained for both PE-140 and PE-120 by sieving. Furthermore, the viscosity-averaged molecular weight tends to increase as the particle size increases for PE-140, i.e. there is a positive correlation between the particle size and the viscosity-averaged molecular weight. However, there is no correlation between particle size and viscosity-averaged molecular weight for PE-120. As mentioned above, the polymer globules in the PE-120 particles are not densely packed, and held together by a large number of polymer fibrils, thus no correlation between particle size and viscosity-averaged molecular weight can be obtained, while the polymer globules in the PE-140 particles are densely packed and a positive correlation between the particle size and the viscosity-averaged molecular weight can be observed.

Next, the effect of particle size on the rheological properties of nascent UHMWPE in LP with the concentrations of 0.1% was further explored, as illustrated in figure 10. It should be noted that the smaller the particle size, the lower the temperature corresponding to the sudden change in viscosity. This is because the nascent UHMWPE with smaller particle sizes in LP melts first, which could be confirmed in figure 6, and the molten polymer chains diffuse into the solvents, causing the sudden increase in the viscosity to occur at a lower temperature compared with the bigger particle sizes. Besides, the viscosity of UHMWPE/LP solution increases as the particle size increases for PE-140. It is well known that the viscosity of UHMWPE/LP solution is closely related to the molecular weight of UHMWPE, and the larger the molecular weight, the higher the viscosity of solution. This result further illustrates that the larger the particle size, the higher the molecular weight of nascent UHMWPE for PE-140. And no correlation between particle size and the viscosity is observed for PE-120.

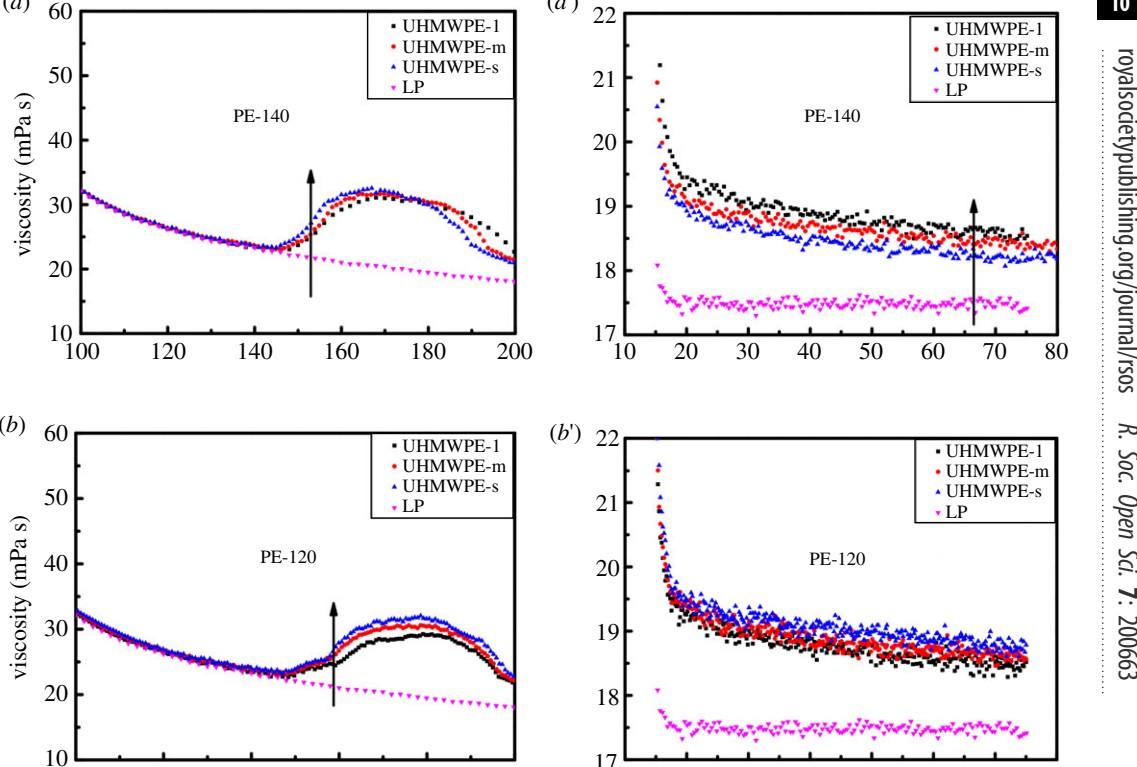

**Figure 10.** Viscosity changes of UHMWPE/LP system with the concentration of 0.1% at a heating rate of 10°C min⁻¹ (a,b) and isothermal process at 200°C (a′,b′).

# 4. Conclusion

The particle morphology, structure and properties of nascent UHMWPE prepared via pre-polymerization technique, and direct polymerization were investigated. Using pre-polymerization technique is an effective way to achieve good particle morphology. The polymer globules in the nascent UHMWPE prepared by using pre-polymerization technique are densely packed and a positive correlation between the particle size and the viscosity-averaged molecular weight can be observed. While the polymer globules in the nascent UHMWPE prepared by direct polymerization are not densely packed and there is no correlation between particle size and viscosity-averaged molecular weight. The nascent UHMWPE prepared by direct polymerization exhibits higher melting temperature and crystallinity, compared with the nascent UHMWPE prepared by using pre-polymerization technique. Besides, the split phenomenon of particles and the fluctuation in the viscosity of UHMWPE/LP system prepared by direct polymerization can be observed at a low heating rate (1°C min⁻¹) due to the presence of a large number of polymer fibrils between the polymer globules. The swelling of nascent UHMWPE particles in LP can be observed when the crystals start to melt at a low heating rate (1°C min⁻¹), while faster heating rate (10°C min⁻¹) would lead to a rapid melting of crystals, and the swelling of UHMWPE particles are hardly observed until the crystals are completely melted.

Data accessibility. The datasets supporting this article have been uploaded as the electronic supplementary material.
Authors' contributions. W.Z. participated in data analysis and the design of the study and drafted the manuscript; Z.W. and H.M. carried out the statistical analyses; X.W. collected field data; J.L., Y.M. and J.Y. conceived of the study, designed the study, coordinated the study and helped draft the manuscript. All authors gave final approval for publication.
Competing interests. We declare we have no competing interests.
Funding. We received no funding for this study.

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
