## [Reviewer comments · Royal Society Open Science]

Review History

RSOS-200663.R0 (Original submission)

Review form: Reviewer 1

Is the manuscript scientifically sound in its present form?

Yes

Are the interpretations and conclusions justified by the results?

Yes

Is the language acceptable?

Yes

Do you have any ethical concerns with this paper?

No

Have you any concerns about statistical analyses in this paper?

No

Recommendation?

Accept with minor revision (please list in comments)

Comments to the Author(s)

The manuscript (RSOS-200663) prepared two nascent UHMWPE particles having different morphologies varying with the polymerization technologies. The formation mechanism of this two nascent particles was discussed and, their crystalline structure (XRD patterns), melting (DSC thermographs), swelling (heating in liquid paraffin), and rheological (rotary rheometer) behaviors were investigated. Some interesting results were obtained in this paper. Overall, the paper has a merit of introducing an effective approach (pre-polymerization technique) to fabricate a nascent UHMWPE powders with regular particle morphologies. Moreover, the manuscript is well written, and the characterizations are well correlated. However, some minor problems (listed as follows) should be addressed before its possible publication.

1. The authors stated, a weak shoulder peak at around $2\theta = 19.5^\circ$ can be assigned to the 001 diffraction of monoclinic phase of PE (Page 11, Para. 1, lines 3-5). However, as the reviewer's knowledge, the response peak should be assigned to (010)M; please check.
2. The crystallinity (XRD) in Table 2 is calculated considering the crystalline fractions [(110)O + (200)O] or [(110)O + (200)O + (010)M]? As the authors stated, The fibril structure stretched by mechanical stress is prone to produce the monoclinic phase, forming the diffraction peak of (001) crystal plane (Page 11, Para. 1, lines 12-13). The reviewer suggests the authors provide the respective fraction of (010)M, (110)O and (200)O in Table 2.
3. The authors stated, the volume of PE-120 particles stays almost constant with the temperature ranging from 30 °C to 130 °C (Page 13, Para. 2, lines 2-3). However, the reviewer observes the significant volume change from 110 to 130 °C in Fig. 5; please check.
4. Please maintain the unity of the effective numbers of the data in Tables 2 and 3.
5. Please provide the scale bar of the SEM images in Fig. 1. Besides, the reviewer feels the information in supplementary material (SI) is repeated with Fig. 1; SI was not mentioned in the text of this manuscript.
6. Please provide the full name of "LP" in the Abstract; the abbreviation LP (liquid paraffin) can be given at its first appearance in the main text, next, use LP only (the readers can get it); the same considerations also applied to "PE-120", "PE-140" in the Conclusion.
7. Please provide the reference (or standard) for the equation to calculate the viscosity-average molecular weight of the samples.
8. As the reviewer's knowledge, the home of the rheometer MCR 302 (Anton Paar) is Austria, please check; alternatively, can the authors convert the rotary speed (300 rpm) into shear rate in the manuscript (Page 7). Note: this is an optional choice.
9. The English expression of some sentences in this manuscript should be polished. Please check the following sentences:
 - (1) Page 7: Optical photographs of UHMWPE particles in liquid paraffin during the heating process were observed a Linkam THMS 600 Stage with Imaging Station (Linkam Scientific Instruments, UK).
 - (2) Page 10: As mentioned in the introduction, the properties of the depend not only on the polymer structure including molecular architecture and crystallization, but also on the morphology of polymer particles.
 - (3) Page 12: Meanwhile, an increase in particle volume originating from the diffusional transport of solvent (LP) into the interior of particles with temperature also can be observed...
10. Please improve the resolution of the insets in Fig. 8; alternatively, the authors can divide the regions in the main images directly; please check the units of ordinates in Figs. 9 and 10.
11. Please maintain the unity of the format of figure caption [With or Without the period (.): Figs. 1, 2, 8, 9 vs. Figs. 3-7, 10].
12. Please add a space between a number and an unit, like 10L (Page 6, Para. 1, line 5) and 1.54Å (Page 6, Para. 4, line 2).
13. Please check the format of refs. 5, 9, 10, 20, 21.

Last but not least, the reviewer is looking forward to read the further/future reports involving the structures and properties of UHMWPE products manufactured using the two nascent powders.

Review form: Reviewer 2

Is the manuscript scientifically sound in its present form?

Yes

Are the interpretations and conclusions justified by the results?

Yes

Is the language acceptable?

Yes

Do you have any ethical concerns with this paper?

No

Have you any concerns about statistical analyses in this paper?

No

Recommendation?

Accept with minor revision (please list in comments)

Comments to the Author(s)

This is an important work dealing with correlations between particle morphology of UHMWPE and its dissolution behavior in paraffin oil. To my knowledge, there are very few literature reports on this topic. The knowledge obtained in this work can promote development of UHMWPE with improved processing properties. The results presented in this article have satisfactory quality. I would support its publication if the following suggestions are adequately addressed:

1. One of the two UHMWPE samples was prepared by the pre-polymerization technique. To better understand the mechanism of morphology improvement by pre-polymerization, it is necessary to analyze particle morphology of the pre-polymer (polymer collected just after the pre-polymerization) by SEM. The particle morphology of PE particles in the initial stage of direct polymerization run (better with similar PE/Cat mass ratio as the prepolymer) can be analyzed for comparison.
2. In Table 3 the PE samples were sieved into three parts with different sizes. It is necessary to list weight fraction of the three parts in order to show the particle size distribution.
3. There are some improper expressions in the text, e.g: P.5-L17, "initial particle growth could be better control" should be "initial particle growth could be better controlled". P.5, "..., forming a large of physical entanglements." should be "..., forming a large number of physical entanglements." P.16 "temperature, and crystals start to melting" should be "temperature, and crystals start to melt".

Decision letter (RSOS-200663.R0)

Dear Dr Zhang:

Title: Particle morphology, structure and properties of nascent UHMWPE
Manuscript ID: RSOS-200663

Thank you for submitting the above manuscript to Royal Society Open Science. On behalf of the Editors and the Royal Society of Chemistry, I am pleased to inform you that your manuscript will be accepted for publication in Royal Society Open Science subject to minor revision in accordance with the referee suggestions. Please find the reviewers' comments at the end of this email.

The reviewers and handling editors have recommended publication, but also suggest some minor revisions to your manuscript. Therefore, I invite you to respond to the comments and revise your manuscript.

Because the schedule for publication is very tight, it is a condition of publication that you submit the revised version of your manuscript before 07-Jun-2020. Please note that the revision deadline will expire at 00.00am on this date. If you do not think you will be able to meet this date please let me know immediately.

Kind regards,
Dr Laura Smith
Publishing Editor, Journals

On behalf of the Subject Editor Professor Anthony Stace and the Associate Editor Dr Chaohua Cui.

RSC Associate Editor:
Comments to the Author:
(There are no comments.)

RSC Subject Editor:
Comments to the Author:
(There are no comments.)

Reviewer comments to Author:
Reviewer: 1

Comments to the Author(s)

The manuscript (RSOS-200663) prepared two nascent UHMWPE particles having different morphologies varying with the polymerization technologies. The formation mechanism of this two nascent particles was discussed and, their crystalline structure (XRD patterns), melting (DSC thermographs), swelling (heating in liquid paraffin), and rheological (rotary rheometer) behaviors were investigated. Some interesting results were obtained in this paper. Overall, the paper has a merit of introducing an effective approach (pre-polymerization technique) to fabricate a nascent UHMWPE powders with regular particle morphologies. Moreover, the manuscript is well written, and the characterizations are well correlated. However, some minor problems (listed as follows) should be addressed before its possible publication.

1. The authors stated, a weak shoulder peak at around $2\theta = 19.5^\circ$ can be assigned to the 001 diffraction of monoclinic phase of PE (Page 11, Para. 1, lines 3-5). However, as the reviewer's knowledge, the response peak should be assigned to (010)M; please check.
2. The crystallinity (XRD) in Table 2 is calculated considering the crystalline fractions [(110)O + (200)O] or [(110)O + (200)O + (010)M]? As the authors stated, The fibril structure stretched by mechanical stress is prone to produce the monoclinic phase, forming the diffraction peak of (001) crystal plane (Page 11, Para. 1, lines 12-13). The reviewer suggests the authors provide the respective fraction of (010)M, (110)O and (200)O in Table 2.
3. The authors stated, the volume of PE-120 particles stays almost constant with the temperature ranging from 30 °C to 130 °C (Page 13, Para. 2, lines 2-3). However, the reviewer observes the significant volume change from 110 to 130 °C in Fig. 5; please check.
4. Please maintain the unity of the effective numbers of the data in Tables 2 and 3.

5. Please provide the scale bar of the SEM images in Fig. 1. Besides, the reviewer feels the information in supplementary material (SI) is repeated with Fig. 1; SI was not mentioned in the text of this manuscript.
 6. Please provide the full name of "LP" in the Abstract; the abbreviation LP (liquid paraffin) can be given at its first appearance in the main text, next, use LP only (the readers can get it); the same considerations also applied to "PE-120", "PE-140" in the Conclusion.
 7. Please provide the reference (or standard) for the equation to calculate the viscosity-average molecular weight of the samples.
 8. As the reviewer's knowledge, the home of the rheometer MCR 302 (Anton Paar) is Austria, please check; alternatively, can the authors convert the rotary speed (300 rpm) into shear rate in the manuscript (Page 7). Note: this is an optional choice.
 9. The English expression of some sentences in this manuscript should be polished. Please check the following sentences:
 - (1) Page 7: Optical photographs of UHMWPE particles in liquid paraffin during the heating process were observed a Linkam THMS 600 Stage with Imaging Station (Linkam Scientific Instruments, UK).
 - (2) Page 10: As mentioned in the introduction, the properties of the depend not only on the polymer structure including molecular architecture and crystallization, but also on the morphology of polymer particles.
 - (3) Page 12: Meanwhile, an increase in particle volume originating from the diffusional transport of solvent (LP) into the interior of particles with temperature also can be observed...
 10. Please improve the resolution of the insets in Fig. 8; alternatively, the authors can divide the regions in the main images directly; please check the units of ordinates in Figs. 9 and 10.
 11. Please maintain the unity of the format of figure caption [With or Without the period (.)]: Figs. 1, 2, 8, 9 vs. Figs. 3-7, 10].
 12. Please add a space between a number and an unit, like 10L (Page 6, Para. 1, line 5) and 1.54Å (Page 6, Para. 4, line 2).
 13. Please check the format of refs. 5, 9, 10, 20, 21.
- Last but not least, the reviewer is looking forward to read the further/future reports involving the structures and properties of UHMWPE products manufactured using the two nascent powders.

Reviewer: 2

Comments to the Author(s)

This is an important work dealing with correlations between particle morphology of UHMWPE and its dissolution behavior in paraffin oil. To my knowledge, there are very few literature reports on this topic. The knowledge obtained in this work can promote development of UHMWPE with improved processing properties. The results presented in this article have satisfactory quality. I would support its publication if the following suggestions are adequately addressed:

1. One of the two UHMWPE samples was prepared by the pre-polymerization technique. To better understand the mechanism of morphology improvement by pre-polymerization, it is necessary to analyze particle morphology of the pre-polymer (polymer collected just after the pre-polymerization) by SEM. The particle morphology of PE particles in the initial stage of direct polymerization run (better with similar PE/Cat mass ratio as the prepolymer) can be analyzed for comparison.
2. In Table 3 the PE samples were sieved into three parts with different sizes. It is necessary to list weight fraction of the three parts in order to show the particle size distribution.
3. There are some improper expressions in the text, e.g: P.5-L17, "initial particle growth could be better control" should be "initial particle growth could be better controlled". P.5, "..., forming a large of physical entanglements." should be "..., forming a large number of physical entanglements." P.16 "temperature, and crystals start to melting" should be "temperature, and crystals start to melt".

Author's Response to Decision Letter for (RSOS-200663.R0)

See Appendix A.

Decision letter (RSOS-200663.R1)

Dear Dr Zhang:

Title: Particle morphology, structure and properties of nascent UHMWPE
Manuscript ID: RSOS-200663.R1

It is a pleasure to accept your manuscript in its current form for publication in Royal Society Open Science. The chemistry content of Royal Society Open Science is published in collaboration with the Royal Society of Chemistry.

On behalf of the Subject Editor Professor Anthony Stace and the Associate Editor Dr Chaohua Cui.

RSC Associate Editor
Comments to the Author:
(There are no comments.)

Reviewer(s)' Comments to Author:

Appendix A

Dear Editor,

Thank you so much for your kind efforts on our manuscript. We appreciated the professional and helpful comments from the reviewers, which are valuable and helpful for us to revise and improve our manuscript and for our further study. We revised the manuscript carefully according to the suggestions from you and the comments of reviewers. The revisions were marked in the new version of the manuscript and the details of the revisions and responses are listed in following.

Reviewer #1:

The manuscript (RSOS-200663) prepared two nascent UHMWPE particles having different morphologies varying with the polymerization technologies. The formation mechanism of this two nascent particles was discussed and, their crystalline structure (XRD patterns), melting (DSC thermographs), swelling (heating in liquid paraffin), and rheological (rotary rheometer) behaviors were investigated. Some interesting results were obtained in this paper. Overall, the paper has a merit of introducing an effective approach (pre-polymerization technique) to fabricate a nascent UHMWPE powders with regular particle morphologies. Moreover, the manuscript is well written, and the characterizations are well correlated. However, some minor problems (listed as follows) should be addressed before its possible publication.

Comment 1: The authors stated, a weak shoulder peak at around $2\theta = 19.5^\circ$ can be assigned to the 001 diffraction of monoclinic phase of PE (Page 11, Para. 1, lines 3-5). However, as the reviewer's knowledge, the response peak should be assigned to (010)M; please check.

Authors' reply 1: Thank you for your suggestion. We made a mistake and the response peak should be assigned to (010)M. The corresponding parts are revised and marked with the blue color in the manuscript.

Comment 2: The crystallinity (XRD) in Table 2 is calculated considering the crystalline fractions [(110)O + (200)O] or [(110)O + (200)O + (010)M]? As the authors stated, the fibril structure stretched by mechanical stress is prone to produce

the monoclinic phase, forming the diffraction peak of (001) crystal plane (Page 11, Para. 1, lines 12-13). The reviewer suggests the authors provide the respective fraction of (010)M, (110)O and (200)O in Table 2.

Authors' reply 2: Thanks for your careful checks. The crystallinity (XRD) in Table 2 is calculated considering the crystalline fractions [(110)O + (200)O + (010)M]. The crystallinity (Xc) was calculated by the ratio of the areas of the crystalline peaks (010), (110) and (200) to the total areas.

Comment 3: The authors stated, the volume of PE-120 particles stays almost constant with the temperature ranging from 30 °C to 130 °C (Page 13, Para. 2, lines 2-3). However, the reviewer observes the significant volume change from 110 to 130 °C in Fig. 5; please check.

Authors' reply 3: Thank you for your question. The UHMWPE particles in LP would move with temperature increasing. In contrast to the latter case, no obvious volume change can be observed with the temperature ranging from 30 °C to 130 °C.

Comment 4: Please maintain the unity of the effective numbers of the data in Tables 2 and 3.

Authors' reply 4: Thanks for your careful checks. The numbers of the data in Tables 2 and 3 are revised and marked with the blue color in the manuscript.

Comment 5: Please provide the scale bar of the SEM images in Fig. 1. Besides, the reviewer feels the information in supplementary material (SI) is repeated with Fig. 1; SI was not mentioned in the text of this manuscript.

Authors' reply 5: The scale bar of the SEM images has been provided in Fig. 1. And the original SEM image are shown in the supplementary material (SI).

Comment 6: Please provide the full name of "LP" in the Abstract; the abbreviation LP (liquid paraffin) can be given at its first appearance in the main text, next, use LP only (the readers can get it); the same considerations also applied to "PE-120", "PE-140" in the Conclusion.

Authors' reply 6: Thanks for your careful checks. The full name of "LP" has been provided in the Abstract. The corresponding parts are revised and marked with the blue color in the manuscript.

Comment 7: Please provide the reference (or standard) for the equation to calculate the viscosity-average molecular weight of the samples.

Authors' reply 7: The intrinsic viscosity $[\eta]$ of UHMWPE in decahydronaphthalene was measured using an Ubbelohde viscometer at 135 °C according to ISO1628-3. The corresponding parts are revised and marked with the blue color in the manuscript.

Comment 8: As the reviewer's knowledge, the home of the rheometer MCR 302 (Anton Paar) is Austria, please check; alternatively, can the authors convert the rotary speed (300 rpm) into shear rate in the manuscript (Page 7). Note: this is an optional choice.

Authors' reply 8: Thanks for your careful checks. The corresponding parts are revised, and the viscosities of UHMWPE/LP system at designed concentrations with temperature were measured by using a rheometer (MCR 302, Anton Paar, Austria). Here, we do not convert the rotary speed (300 rpm) into shear rate in the manuscript because the rotation speed is the setting condition of our experiment.

Comment 9: The English expression of some sentences in this manuscript should be polished. Please check the following sentences:

(1) Page 7: Optical photographs of UHMWPE particles in liquid paraffin during the heating process were observed a Linkam THMS 600 Stage with Imaging Station (Linkam Scientific Instruments, UK).

(2) Page 10: As mentioned in the introduction, the properties of the depend not only on the polymer structure including molecular architecture and crystallization, but also on the morphology of polymer particles.

(3) Page 12: Meanwhile, an increase in particle volume originating from the diffusional transport of solvent (LP) into the interior of particles with temperature also can be observed...

Authors' reply 9: Thanks for your professional and helpful suggestions.

(1) Page 7: Optical photographs of UHMWPE particles in liquid paraffin during the heating process were observed by a Linkam THMS 600 Stage with Imaging Station (Linkam Scientific Instruments, UK);

(2) Page 10: As mentioned in the introduction, the properties of polymer depend not only on the polymer structure including molecular architecture and crystallization, but also on the morphology of polymer particles.

(3) Page 12: Meanwhile, an increase in particle volume with temperature originating from the diffusional transport of solvent (LP) into the interior of particles can also be observed.

The corresponding parts are revised and marked with the blue color in the manuscript.

Comment 10: Please improve the resolution of the insets in Fig. 8; alternatively, the authors can divide the regions in the main images directly; please check the units of ordinates in Figs. 9 and 10.

Authors' reply 10: The DPI resolution of the insets in Fig. 8 is 600. However, the scale of the insets is small. Thus, the scale of the figures is enlarged in Fig. 8. And the units of ordinates in Figs. 9 and 10 are right.

Comment 11: Please maintain the unity of the format of figure caption [With or Without the period (.): Figs. 1, 2, 8, 9 vs. Figs. 3-7, 10].

Authors' reply 11: Thanks for your professional and helpful suggestions. The format of figure caption has been united.

Comment 12: Please add a space between a number and an unit, like 10L (Page 6, Para. 1, line 5) and 1.54Å (Page 6, Para. 4, line 2).

Authors' reply 12: Thanks for your professional and helpful suggestions. The space between a number and an unit has been added.

Comment 13: Please check the format of refs. 5, 9, 10, 20, 21.

Authors' reply 13: Thanks for your professional and helpful suggestions. The format of refs. 20, 21 has been revised. However, the doi could not find in refs. 5, 9, 10, because the references are old.

Comment 14: Last but not least, the reviewer is looking forward to read the further/future reports involving the structures and properties of UHMWPE products manufactured using the two nascent powders.

Authors' reply 14: In the next step we will make the finished product such as fibers and explore the relationship between the structure and performance of the finished product.

Reviewer #2:

Comments: This is an important work dealing with correlations between particle morphology of UHMWPE and its dissolution behavior in paraffin oil. To my knowledge, there are very few literature reports on this topic. The knowledge obtained in this work can promote development of UHMWPE with improved processing properties. The results presented in this article have satisfactory quality. I would support its publication if the following suggestions are adequately addressed:

Comment 1: One of the two UHMWPE samples was prepared by the pre-polymerization technique. To better understand the mechanism of morphology improvement by pre-polymerization, it is necessary to analyze particle morphology of the pre-polymer (polymer collected just after the pre-polymerization) by SEM. The particle morphology of PE particles in the initial stage of direct polymerization run (better with similar PE/Cat mass ratio as the prepolymer) can be analyzed for comparison.

Authors' reply 1: Thanks for your professional and helpful suggestions. Next, we will continue to consider the formation mechanism of the particle morphology by the pre-polymerization technology.

Comment 2: In Table 3 the PE samples were sieved into three parts with different sizes. It is necessary to list weight fraction of the three parts in order to show the particle size distribution.

Authors' reply 2: Thank you for your suggestion. In order to explore the relationship between the viscosity-averaged molecular weight and particle size, the PE samples were sieved into three parts with different sizes. In our opinion, the weight fraction of the three parts is not important.

Comment 3: There are some improper expressions in the text, e.g: P.5-L17, "initial particle growth could be better control" should be "initial particle growth could be

better controlled". P.5, "..., forming a large of physical entanglements." should be "..., forming a large number of physical entanglements." P.16 "temperature, and crystals start to melting" should be "temperature, and crystals start to melt".

Authors' reply 3: Thank you for your suggestion. The improper expressions have been revised and marked with the blue color in the manuscript.

Thanks again for the editor's and reviewers' professional and helpful suggestions and comments to our manuscript. We wish our responses would be helpful for the editor to make a decision for publication of our manuscript in *Royal Society Open Science*.

We wish the improved manuscript would be acceptable for publication.

Sincerely yours,

Wenyang Zhang and coauthors